# A Scoping Review on the Effects of Carotenoids and Flavonoids on Skin Damage Due to Ultraviolet Radiation

**DOI:** 10.3390/nu15010092

**Published:** 2022-12-24

**Authors:** Kirushmita Anbualakan, Nur Qisti Tajul Urus, Suzana Makpol, Adawiyah Jamil, Elvy Suhana Mohd Ramli, Suria Hayati Md Pauzi, Norliza Muhammad

**Affiliations:** 1Department of Pharmacology, Faculty of Medicine, Universiti Kebangsaan Malaysia, Kuala Lumpur 56000, Malaysia; 2Department of Biochemistry, Faculty of Medicine, Universiti Kebangsaan Malaysia, Kuala Lumpur 56000, Malaysia; 3Department of Medicine, Faculty of Medicine, Universiti Kebangsaan Malaysia, Kuala Lumpur 56000, Malaysia; 4Department of Anatomy, Faculty of Medicine, Universiti Kebangsaan Malaysia, Kuala Lumpur 56000, Malaysia; 5Department of Pathology, Faculty of Medicine, Universiti Kebangsaan Malaysia, Kuala Lumpur 56000, Malaysia

**Keywords:** phytonutrients, ultraviolet, photodamage, carotenoids, flavonoids, ageing

## Abstract

Skin exposure to ultraviolet (UV) rays in the sun causes premature ageing and may predispose to skin cancers. UV radiation generates excessive free radical species, resulting in oxidative stress, which is responsible for cellular and DNA damage. There is growing evidence that phytonutrients such as flavonoids and carotenoids may impede oxidative stress and prevent photodamage. We conducted a systematic review of the literature to explore the effects of certain phytonutrients in preventing skin photodamage. We searched the electronic Medline (Ovid) and Pubmed databases for relevant studies published between 2002 and 2022. The main inclusion criteria were articles written in English, and studies reporting the effects of phytonutrient-containing plants of interest on the skin or skin cells exposed to UV radiation. We focused on tea, blueberries, lemon, carrot, tomato, and grapes, which are rich in flavonoids and/or carotenoids. Out of 434 articles retrieved, 40 were identified as potentially relevant. Based on our inclusion criteria, nine articles were included in the review. The review comprises three combined in vitro and animal studies, four human studies, one in vitro research, and one mixed in vitro and human study. All the studies reported positive effects of flavonoids and carotenoid-containing plant extract on UV-induced skin damage. This evidence-based review highlights the potential use of flavonoids and carotenoids found in plants in preventing the deleterious effects of UV radiation on the skin. These compounds may have a role in clinical and aesthetic applications for the prevention and treatment of sunburn and photoaging, and may potentially be used against UV-related skin cancers.

## 1. Introduction

Exposure to ultraviolet (UV) rays not only causes unsightly skin conditions, such as wrinkles, redness, rough skin and pigmentation, but is also responsible for the development of skin cancers [1]. Reactive oxygen species (ROS), which are produced when riboflavin, porphyrins, and heme-containing proteins absorb photons from UV radiation, are the primary mechanism by which UV radiation harms the skin [2]. ROS are unstable metabolic by-products of oxygen molecules produced in biological systems. Examples of ROS molecules include superoxide radicals, hydroxyl radicals, hydrogen peroxides, lipid peroxides and singlet oxygen [3]. Depending on their concentration, these molecules exhibit both harmful and beneficial effects in organisms [4]. ROS play a beneficial role in various biological processes when maintained at low or moderate levels, such as in cell proliferation, signal transduction, gene expression, and host defence [5]. A balanced level of ROS is necessary for the modulation of proper signalling mechanisms, as in apoptotic pathways, cell regulation and phagocytosis [6]. Excessive production of ROS leads to oxidative stress, an alarming condition that results in harmful cellular metabolic consequences due to oxidative damage to DNA, proteins, lipids and other biomolecules [7].

Phytonutrients are natural chemicals derived from plants. They are often used as dietary supplements and are recognised as foods. There are thousands of phytochemicals found in plants which include carotenoids, flavonoids, phytoestrogens and resveratrol [8]. With the advancement in agricultural technology, phytonutrients can be mass-produced, easily and are conveniently accessible, and they are widely used all over the world to promote health and well-being. Their usage is becoming increasingly popular as more scientific discoveries are made to unearth their benefits. Phytonutrients exhibit favourable properties, mainly as antioxidants, scavenging free radicals to overcome oxidative stress [9]. In addition, they also manifest anti-bacterial [10], anti-fungal [11], anti-inflammatory [12], anti-allergic [13], anti-spasmodic & hypolipidemic [14], neuroprotective [15], and antihypertensive [16] properties. Some phytonutrients induce cellular apoptosis and thus have potential chemo-preventive qualities, some act as diuretics and analgesics, and many others display carminative and immunomodulatory effects [17,18]. Various parts of plants, such as roots, leaves, flowers and seeds, have been used traditionally to prevent and cure ailments. The leaves and flowers are often consumed fresh, as in salads, or dried and later steeped in boiling water to be served as teas. The roots, or sometimes the whole plants, either dried or fresh, are boiled for long hours before the concoctions are ready to be sipped. The seeds are often extracted (e.g., cold-pressed) for oils [8]. Besides oral consumption, the plant parts can also be processed for topical application, especially for the treatment of skin disorders, as ointments for muscular pains, lotions, creams and facial masks to prevent wrinkles and other facial conditions. Commercially available phytonutrients are either sold as purified compounds, which are more expensive because of their higher manufacturing costs, or sold as dried crude extracts, which are frequently combined with other vitamins and dietary supplements in capsules or other formulations. In this review, we examine data from the literature on the effects of certain phytonutrients against skin damage due to UV irradiation. We chose flavonoids and carotenoids found in tea, carrots, blueberries, lemons, grapes, and tomatoes. These plants are of special interests as people across the globe widely consume them.

## 2. Materials and Methods

### 2.1. Literature Review

The Preferred Reporting Items for Systematic Reviews and Meta-Analyses (PRISMA) standards for a scoping review were followed in this systematic review (Table 1). We used Pubmed (published between 2002 and 2022) and Medline via Ovid Medline (published between 1946 and week 4 of October 2022) to conduct a thorough search of health science journals. The search strategy involved a combination of the following two sets of keywords: (1) tea OR blueberr* OR lemon OR carrot OR tomato* OR grape* AND (2) skin OR ageing OR aging OR photoag* OR UV* OR ultraviolet* OR sun* OR wrinkl*.

### 2.2. Selection of Research Article

Results were restricted to research that was written in English. Studies with the following features were incorporated: (1) the effects of flavonoids and carotenoid-containing plants on skin damage due to ultraviolet radiation, and (2) skin damage or wrinkles should be related to sun exposure or experimentally induced under UV influence. Review papers, news pieces, letters, editorials, and case studies were not included in this systematic review.

### 2.3. Data Extraction and Management

Before being included in the review, articles underwent three stages of screening. In the initial phase, any publications that did not meet the criteria for inclusion based solely on the title were disqualified. The remaining articles’ abstracts were scrutinized in the second phase, and those that did not fulfil our inclusion requirements were disqualified. In the final phase, full-text articles were acquired and carefully examined to filter out any that did not meet our inclusion criteria. After duplicates were eliminated, the remaining papers underwent a second round of screening by at least two reviewers. Before moving on to the data extraction stage, the full articles needed to be approved by both reviewers. Any differences in opinions were resolved through discussion between the reviewers. All data extraction was done independently using a data collection form to standardize the data collection process. The following information was gathered from the studies: (1) the type of study, (2) the kind of extract or sample used in the study, (3) a brief description of the sample/population of the study, (4) a brief description of the procedures used in the investigation, and (5) a summary of the study’s results.

## 3. Results

### 3.1. Search Results

Forty potentially pertinent papers were found from the literature searches. Based on the title and abstract, two reviewers separately decided whether or not to include or exclude each article. Nine publications in total were retrieved for further analysis and data extraction. The remaining publications were either omitted because they lacked primary investigations or because they were not related to phytonutrients and UV skin damage. Figure 1 depicts a flow chart of the selection of paper process, along with the reasons for exclusion.

### 3.2. Study Characteristic

Table 1 presents a summary of the characteristics of all investigations. All the studies were published between the years 2002 to 2022 and consisted of three combined animal and in vitro studies, four human studies, one in vitro study and one combined in vitro and human study. Mice were employed as the experimental model in all animal investigations, in which animals’ skin was exposed to UVB radiation for roughly seven to fifteen weeks. As for the in vitro studies, human fibroblast HS68, human dermal fibroblast, human keratinocytes cell, and HaCaT human keratinocyte cells were cultured. In the human study, various skin phototypes of healthy people were used, and the volunteers’ skin was exposed to UVB radiation for three days to twelve weeks. All studies used an experimental design that compared the results of the UV-exposed groups treated with fruit extract to the control or sham groups.

### 3.3. Findings from In Vitro Studies

UVB treatment of HS68 cells greatly increased protein oxidation. Green tea polyphenols (GTP) inhibited the UVB-induced protein oxidation of HS68 cells in a dose dependent manner [19]. In another study on fibroblast cells, UV-B radiation at 100 mJ/cm^2^ lowered cell viability by 30% and caused cell death. Treatment of UVB-irradiated cells with anthocyanin-rich extract from bog blueberry (ATH-BBe) at concentrations between 1 and 10 mg/L increased cell viability dose-dependently. UV-B exposure significantly reduced procollagen expression. By treating the fibroblast cells with ATH-BBe, the depleted procollagen expression was elevated in a dose-dependent manner. Additionally, UV-B irradiation significantly increased the secretion of MMP-1, MMP-8, and MMP-13. Treatment with 1–10 mg/L ATH–BBe inhibited the production of all three enzymes in a dose-dependent manner. UV-B also increased cytokines release, specifically IL-1b, IL-6, IL-8, and TNF-a. The ATH-BBe treatment inhibited these cytokines secretion. Furthermore, phosphorylation of IkB in human dermal fibroblasts was activated by UV-B radiation, leading to NF-kB activation and MAPK signalling. It was noted that pre-treatment with ATH-BBe decreased IkB phosphorylation dose-dependently [21].

Perez-Sanchez et al. (2014) observed an increase in the survival of HaCaT cells after UVB irradiation following administration of flavonoid-rich mixture of rosemary and citrus extracts, suggesting possible synergistic effects. The extract combination also significantly reduced chromosomal abnormalities as well as intracellular reactive oxygen species (ROS) and DNA damage in HaCaT cells [22]. Luteolin is a natural flavonoid found in carrots, and pretreatment with luteolin significantly and dose-dependently reduced UVB-induced MMP-1 expression. However, luteolin did not prevent the activation of JNK, ERK, and p90RSK caused by UVB exposure. Luteolin pretreatment inhibited UVB-induced AP-1 transactivation and reduced UVB-induced c-Fos gene expression in HaCaT cells. Luteolin significantly reduced the activity of JNK1 and p90RSK, but not JNK2 and ERK2. Luteolin bound to p90RSK2 in an ATP-independent manner and JNK1 in an ATP-competitive manner [23].

Kim et al. (2019) conducted a study using grape peel extract (GPE) on irradiated HaCaT cells. It was noted that in the normal control group, the expressions of nuclear Nrf2 and cytoplasmic HO-1 were elevated by UVB irradiation. In the cells that were pretreated with GPE at a concentration of 625 g/mL, the expressions were significantly increased. This suggests that GPE pretreatment was likely to activate the intracellular Nrf2-mediated antioxidant defense system in HaCaT cells, thereby protecting the cells from UV-induced cellular damage [26].

### 3.4. Findings from In Vivo Studies

In the study by Vayalil et al. (2004) on SKH-1 hairless mice, the epidermis of healthy, non-UV exposed skin was thin, with a thickness of two to three cell layers. The skin of mice that received repeated UVB exposure for two months was six to eight cell layers thick, indicating a hyperplastic response. Oral administration of GTP to these UVB-exposed mice resulted in a 67% reduction of the epidermal hyperplastic response. They also had a significant reduction of skin protein carbonylation (by 50%) after UVB exposure. Furthermore, chronic UVB exposure on the mouse skin increased the levels of MMP-2 (3-fold), MMP3 (10-fold), MMP-7 (2-fold), and MMP-9 (2-fold). GTP supplementation reduced the expression of MMP-2, MMP-3, MMP-7, and MMP-9 by 67, 62, 63, and 60%, respectively. In addition, compared to the epidermis, the dermis had higher amounts of protein oxidation due to continuous UVB radiation. Protein oxidation caused by UVB in the dermis was prevented by oral consumption of GTP in drinking water [19].

Lim et al. (2013) administered topical luteolin on SKH-1 mouse skin 1 h before UVB exposure. They reported that luteolin prevented UVB-induced wrinkle development and reduced the overexpression of MMP-13 in SKH-1 mice [23]. Meanwhile, in another study, oral GPE treatment three times per week for seven weeks resulted in a reduction of number of wrinkles and overall wrinkle lengths on mouse skin compared to the UVB-exposed control group. Additionally, pre-treatment with GPE and resveratrol significantly reduced UVB-induced thickening of the epidermis. The nuclear level of Nrf2 was considerably reduced in the epidermis and hepatic tissues of mice exposed to UVB radiation. The simultaneous administration of either GPE or resveratrol caused a significant, dose-dependent increase of Nrf2 level in the nucleus. Additionally, GPE and resveratrol increased the cytosolic level of HO-1, a significant Nrf2 downstream antioxidant enzyme, similar to the way in which the nuclear Nrf2 level was increased [26].

### 3.5. Findings from Human Studies

Heinrich et al. (2011) conducted a randomized double-blind, placebo-controlled study on 60 female volunteers using a beverage containing green tea polyphenols for 12 weeks. Despite UV exposure, the subjects receiving GTP had significantly higher serum EGCG, ECG, epicatechin and catechins levels at weeks 6 and 12 compared to the control group. The skin of the ladies in the intervention group showed less erythema at weeks 6 and 12. At week 12, viscoelasticity was reduced, while biological elasticity and skin density were increased. Skin thinning was unaffected. Surface evaluation at week 12 revealed a significant decrease in skin roughness, volume, and scaling compared to the control group. Skin hydration, dermal blood flow and oxygen saturation were increased in the GTP group, whereas the control group showed no discernible changes [20]. Meanwhile, eight weeks of daily oral administration of the combined rosemary and citrus extract at a dose of 250 mg resulted in a considerable increase in the minimum erythema dose (MED). The photoprotection of this flavonoid-rich mixture was enhanced after 12 weeks of oral consumption [22].

In another randomized, double-blind placebo-controlled study using carotenoid-rich tomato nutrient complex (TNC), Groten et al. (2019) observed that the intervention group had significantly lesser UVB-induced erythema formation compared to the placebo group. The active supplement group also had lower IL6 and TNF levels, with a considerably increase in the plasma carotenoid levels [24]. A similar photoprotective effect of tomato was also noted by Rizwan et al. (2010) who conducted a single-blind, randomized controlled study on 20 white female volunteers. They were given 55 g of tomato paste daily on white bread for 12 weeks, while the control group received 10 mg of olive oil. The intervention group had a significantly lesser degree of redness measured on erythema meter post-treatment compared to before the treatment commenced. MMP-1 was elevated, and fibrillin was decreased prior to UV radiation. Following supplementation, tomato paste reduced UVR-induced MMP-1 compared to control while abrogating UVR-induced fibrillin-1 reduction. The tomato paste treatment was also associated with an increase in procollagen deposition [25].

In the study by Yuan et al. (2012), grape seed proanthocyanidin extract (GSPE) was applied topically and skin biopsies were taken. Compared to the UV group, the GSPE group had fewer mutant p53-positive cells. Most Langerhans cells exhibit longer and more numerous dendrites when GSPE is present, which may indicate immunological activation [27].

## 4. Discussion

The skin, being the largest organ in the body, accounts for roughly 16% of the total body mass. The two main layers of skin are the epidermis and dermis, which together contain epithelial, mesenchymal, glandular, and neurovascular components [28]. Keratinocytes are the most abundant cells in the epidermis. They are arranged in multiple continuous layers, with each layer exhibiting different morphologies and chemical characteristics. They are distinguished from the rest of the skin cells by the presence of intermediate filament proteins called keratins, the development of desmosomes, and tight connections with one another to form a powerful physicochemical barrier [29]. Between the keratinocytes in the basal layers are dendritic-shaped, melanin-producing cells called melanocytes. Melanin is a UV-absorbing pigment distributed to the basal cells via melanocyte dendrites to protect the skin from UV damage [30].

The primary energy source and component necessary for the existence of human is sunlight. On the surface of the planet, ultraviolet radiation makes up 5% of all incident sunlight. Despite having a much lower fraction, it has the highest quantum of energy when compared to other radiations, and therefore it is extremely harmful to human skin [31]. The electromagnetic spectrum contains UV photons, which have a wavelength in between visible light and gamma radiation. According to electrophysical properties, UV energy can be separated into UV-A, UV-B, and UV-C components. UV-C photons have the shortest wavelengths (100–280 nm) and maximum energies, whereas UV-A photons have the longest (315–400 nm) and lowest energies; UV-B photons are in the middle [32]. Between 90–95% of the UV radiation to which we are exposed is UVA, while the remaining comes from UVB. UVC light is blocked by the ozone layer in the atmosphere and therefore does not reach the surface of the earth. Unlike UVB, which is mostly absorbed by the epidermal layer, UVA can penetrate both the epidermis and the deeper dermal layers of the skin [33]. Exposure to UVB is most often associated with sunburn and skin cancers, while UVA exposure is linked to premature skin ageing manifested by displeasing appearances such as wrinkles, pigmentation, rough skin texture and lentigines. UVA also has a role in the development of skin carcinomas by causing indirect DNA damage [34].

Through several mechanisms intimately related to oxidative stress and the generation of ROS, UV radiation damages the skin and speeds up the ageing process. ROS destroy other cellular molecules to promote a series of reactions that result in DNA damage and lipid peroxidation, as well as structural and enzymatic alterations of proteins [35]. Studies have demonstrated that ROS promotes phosphorylation of protein kinases in the extracellular signal-regulated kinase (ERK), c-Jun amino (N)-terminal kinase (JNK1/2/3) and p38 isoform signalling cascade of mitogen-activated protein kinases (MAPK) [36]. MAPK signalling enhances the activity of the activator protein-1 (AP-1) transcription factor, which leads to the activation of transcription factors that activate stress-inducible genes, notably matrix metalloproteinases (MMPs) [37]. Activation of MMPs further leads to the degradation of extracellular matrix (ECM) components in the skin, such as collagen, which provides tensile strength to the dermis. These disrupted collagen products down-regulate new collagen synthesis [38]. Since collagen is an integral structural component of the skin, its loss promotes sagging and the appearance of aged skin, as well as decreased skin hydration [36]. Additionally, the activation of inflammatory mediators by ROS, including interleukin-1, tumour necrosis factor-alpha, and epidermal growth factor (EGRF), results in a reduction in the amount of ECM proteins by elevating MMP or cyclooxygenase-2 while reducing the amount of collagen precursor, procollagen [39].

In this systematic review, we focused on selected flavonoids and carotenoid-rich plants, as they are readily available and widely consumed. Tomatoes, lemons, and carrots are the three plants generally used in cooking or eaten raw in salads. Blueberries and grapes are primarily produced in temperate countries, but they are exported all over the world, and are accessible even to those in tropical climates. Tea, either black or green, is a popular beverage consumed all over the globe. These plants are gaining attention for their positive effects on well-being, and may potentially play a role in functional medicine [40].

Flavonoids are a family of over 5000 compounds found in plants. The flavone nucleus, which has two benzene rings separated by an oxygen-containing pyrene ring, serves as the basis for the structure of a flavonoid. Flavonoids serve a variety of roles in plants, including pigment functions, stress resistance, signalling molecules, and reproductive regulators. They provide ultraviolet protection by blocking harmful exposure of light to the plant interior [41]. Isorhamnetin, catechin, kaempferol, epicatechin, quercetin, and epigallocatechin-3-gallate (EGCG) are the main dietary flavonoids. The ability to prevent DNA damage and influence cellular signalling pathways are among the properties that flavonoids possess. Others include scavenging free radicals, as UV absorbents, and as cytoprotective, anti-inflammatory, and anti-apoptotic factors [42]. Prior studies have shown that foods high in polyphenols (a group which flavonoids are a part of) are effective photoprotectants against UV carcinogenesis [43].

As shown in Table 1, the study by Vayalil et al. [19] demonstrated the anti-ageing and photoprotective role of green tea polyphenols (GTP) in animal subjects. GTP mitigates oxidative stress by preventing oxidative damage to protein macromolecules and inhibiting of expression of MMP, which breaks down extracellular matrix proteins [19]. This finding was further confirmed clinically by Heinrich et al. [20] in a double-blinded, placebo-controlled trial study. It was found that routine consumption of GTP beverages for 12 weeks provided photoprotection against harmful UV radiation. UV-induced erythema was considerably lowered by 25%, while the skin of the women receiving GTP was noted to have significantly better structural parameters such as hydration, elasticity and density. Consuming GTP beverages for 12 weeks boosted skin blood flow and oxygen supply [20].

In addition to oral administration, antioxidant compounds derived from GTP were reported to have beneficial skin effects when applied topically. An example of such a compound is epigallocatechin-3-gallate (EGCG), one of the most prominent tea flavanols. Topical application of EGCG dramatically decreased the production of skin cancers induced by UV irradiation [44]. Moreover, topical EGCG therapy for 30 min before UVA exposure significantly reduced the development of sunburn cells and dermo-epidermal activation compared to exposed, untreated Wistar rats [45]. The administration of other flavonoids topically also positively impacts the skin. Wang et al. (2020) reported that topical application of flavonoids extracted from lemon peel prevented photodamage of UVB-induced skin damage in mice, reduced oxidative stress and mitigated inflammatory reactions responsible for photoaging [46].

The beneficial impacts of flavonoids on UV-induced skin damage were also observed in studies using blueberries and carrots. Bog blueberry extract (BBe) modulates NF-kB-responsive and MAPK-dependent pathways in human dermal fibroblast to suppress UV-B-induced collagen degradation and inflammatory responses [21]. On the other hand, luteolin, a type of flavonoid found in carrots, was noted to reduce UVB-induced MMP-1 production [23]. In addition, topical quercetin and silymarin therapy were found to be protective as well [47].

The minimal erythema dose (MED) is the dose of UV radiation required to elicit redness or erythema in the skin. The higher the MED value, the more radiation energy is needed to cause skin discoloration. In the study by Pérez-Sánchez et al. (2014) conducted on human volunteers, eight weeks of consumption of a beverage containing a blend of citrus flavonoids and rosemary polyphenols produced a substantial increase in the MED. This study demonstrates the photoprotective properties of the polyphenol mixtures, as a larger dose of UV radiation was required, compared to the control participants, to generate similar skin redness [22]. MED determination in animal and human studies involving UV radiation indicates the effectiveness of the compound of interest. The increase in MED was also seen in another double blind, placebo-controlled clinical trial using a mixture of carotenoids [48].

Carotenoids comprise a class of naturally occurring tetraterpenoids, with β-carotene as the most prominent. They are abundant in various fruits and vegetables such as grapes, carrots and tomatoes. The vibrant yellow, orange and red hues of fruits and vegetables, as well as the colour of leaves in the fall after the loss of the covering chlorophyll, are caused by carotenoids [49]. In photosynthetic systems, carotenoids serve as auxiliary components in the light-harvesting complex and offer photoprotection against excessive light exposure and photooxidative damage. The carotenoids α-carotene, β-carotene, lycopene, lutein, and zeaxanthin have been the subject of much research [50]. It is widely known that ROS plays a crucial role in UV-induced oxidative damage [51]. Carotenoids could protect skin against UV ray damage in several ways due to their structure and physicochemical characteristics, including by increasing optical density, quenching singlet oxygen (^1^O_2_), or, in the case of provitamin A carotenoids, forming retinoic acid. Other reactive oxygen species, such as superoxide anions, hydroxyl radicals, and hydrogen peroxide, can also be scavenged by carotenoids [52].

Table 1 summarizes the photoprotective effects of selected carotenoids. In the randomised, double-blind, placebo-controlled study by Groten et al., (2019), supplementing participants with a carotenoid mixture of tomato and rosemary extract before UV-irradiation resulted in a reduction of erythema, and prevented the increase of pro-inflammatory cytokines [24]. Similarly, Rizwan et al. (2011) found that lycopene-rich tomato paste offers defense against the short-term, and possibly long-term, effects of photodamage [25]. Another group of researchers [53] observed that daily consumption of 40g of paste for ten weeks resulted in a 40% decrease in the development of cutaneous erythema brought on by exposure to solar-simulating UV radiation. In addition to tomato, beneficial impacts of carotenoids on UV-induced skin damage were also observed in studies using grape peel extract (GPE). GPE inhibited metalloproteinases and induced Nrf2-dependent antioxidant enzymes, such as heme oxygenase-1 (HO-1) [26]. In addition, GPE reportedly reduced the development of mutant p53 epidermal cells [27].

A growing number of studies have elucidated the cellular and molecular photoprotective mechanisms of polyphenols. In an in vitro study using human keratinocytes, polyphenols were noted to increase cell survival, prevent ROS, and limit DNA damage [46]. In addition, the flavanols quercetin and kaempferol reduced UVA-induced MMP-1 (collagenase) activity and expression in human skin fibroblasts. EGCG was found to have the capability to inhibit MMP-2 and -9 activities [54]. When treated with EGCG, adult human skin fibroblasts and healthy human epidermal keratinocytes that underwent UVA irradiation had significantly less DNA damage [55]. In an in vivo study using the transparent, free-living nematode *C. elegans*, carotenoids derived from *H. lacustris* algae prevented oxidative stress by upregulating the expression of antioxidant enzymes superoxide dismutase (SOD) and catalase [56]. Trivedi & Jena (2015) examined the function of dietary beta-carotene and lipid peroxidation products in activating MMP-9. For the duration of this 8-week experiment, hairless mice were periodically exposed to UVA radiation that caused a considerable rise in the quantity of peroxidized cholesterol in the skin, an increase in MMP-9 protein levels, activity, and wrinkling and sagging formation. When beta-carotene was added to the mice’s food throughout the radiation period, the amount of peroxidized cholesterol was reduced. MMP-9 activity and expression were also reduced, along with the appearance of wrinkles and drooping skin [57]. Figure 2 summarizes the cellular and molecular responses these phytonutrients act upon preventing photodamage.

The pharmacokinetic properties of any phytonutrients should be considered before they are used clinically. To achieve optimum therapeutic efficacy, the transport of chemical compounds and their metabolites from the gut to the blood stream is crucial. Evidence has shown that different types of flavonoids are absorbed at various rates and sites. Hesperetin, naringenin, and eriodictyol are a few examples of the flavanones that are well absorbed from the gastrointestinal tract [58,59,60], while oral quercetin absorption is close to 20% [61]. Other orally absorbed flavonoids include genistein [62], narirutin [63], luteolin [64], flavan-3-ols ((-)-epigallocatechin—EGC- and (-)-epicatechin -EC-) [65], and resveratrol [66]. The absorption and bioavailability of different flavonoids are significantly influenced by food and the position of flavonoids in dietary sources. Evidently, the ethanol in red wine increases the gut’s ability to absorb anthocyanins [67]. When a flavonoid such as quercetin is administered along with carbohydrates, intestinal absorption and bioavailability are both increased [68].

Carotenoids undergo complicated absorption and distribution processes. For optimal absorption, a number of processes must take place [69]. These include: (1) sufficient food digestion to release carotenoids; (2) formation of lipid micelles in the small intestine; (3) carotenoids being taken up by intestinal mucosal cells, and (4) transport of carotenoids or their metabolic products to the lymphatic or portal circulation [70]. There is evidence that bioavailability of carotenoids depends on how they are processed. Heat treatment or mechanical homogenization increases their bioavailability several-fold, while lipid soluble compounds reduce their absorption [71]. Bioavailability of β-carotene is higher when spinach is taken in liquefied form compared to whole leaf spinach [72].

Toxicity is not an issue for flavonoids when they are ingested in concentrations found in foods. However, if they are taken in extremely high doses as supplements, flavonoid poisoning is a possibility. Furthermore, because flavonoids have the capacity to bind nonheme iron, they may be harmful to people with mild iron deficiency, such as the elderly [73]. On the other hand, carotenoids are typically regarded as non-toxic, even when consumed in large concentrations as purified supplements, except under certain conditions. Large dosages of canthaxanthin can cause reversible retinopathy [74]. Additionally, smokers should avoid consuming high amounts of B-carotene because it has been linked to an elevated risk of stomach and lung cancer in some studies [75].

## 5. Conclusions

Flavonoids and carotenoids are phytonutrients that have shown positive results in reducing and preventing skin damage due to UV exposure. It can be deduced that their role in skin protection is via several mechanisms, mostly involving the MAPK, Nrf2 and NF-kB pathways responsible for oxidative stress and inflammatory reactions. Understanding the cellular and molecular mechanisms of photoprotection of these phytonutrients will enable researchers to develop therapeutic agents against UV-related skin disorders, particularly malignancies.

## Figures and Tables

**Figure 1 nutrients-15-00092-f001:**
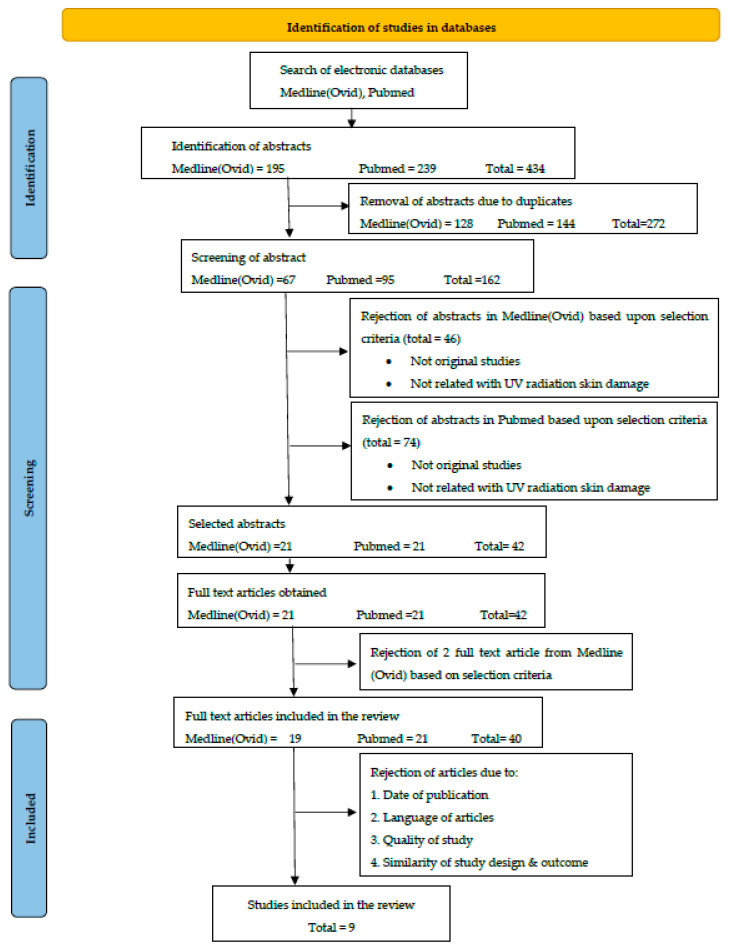
Flow chart of the selection process of the articles in this review.

**Figure 2 nutrients-15-00092-f002:**
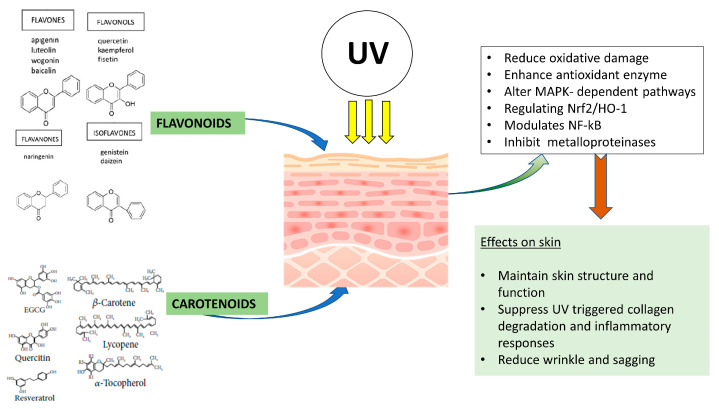
Possible mechanisms of effects of carotenoids and flavonoids on UV damaged skin.

**Table 1 nutrients-15-00092-t001:** Characteristics of the studies included in this review (changes in parameters compared to the normal control).

Type of Plant Used	Sample/ Population	Methodology	Parameters Increased	Parameters Decreased	Parameters Unchanged	Conclusion
Green tea polyphenol (GTP) [19]	In vivo study Female SKH-1 hairless mice (6–7 weeks old).	GTP was given to SKH-1 hairless mice as drinking water (0.2%, wt/vol) before they were repeatedly exposed to UVB (90 mJ/cm^2^) for two months on alternate days. (a) H&E staining for microscopic histological evaluation (b) Skin biopsies were collected to determine protein carbonyl following DNPH analytical assay (c) Western blot analysis was done to assess the protein oxidation & MMP analysis		epidermal hyperplasia protein oxidation protein carbonylation expression of MMP-2, MMP-3, MMP-7, and MMP-9		GTP exerts photoprotective effects by preventing oxidative damage and inhibition of MMP expression.
	In vitro study Human fibroblast HS68.	Human fibroblast HS68 were cultured and treated with GTP before being exposed to UVB radiation (30 mJ per cm^2^) (a) Western blot analysis to access protein oxidation				
Green tea [20]	Double blind, placebo controlled clinicaltrial	(a) UV irradiation sensitivity was tested at weeks 0, 6, and 12 where dorsal skin at the back and scapular region was exposed to radiation. Skin tone was assessed before and after 24 h. (b) Skin elasticity was assessed using the Cutometer SEM 474 suction method on the inner forearm skin. (d) Skin density, thickness, structure and texture were examined using a high-frequency Ultrasound B-Scan. (e) Tewameter and Corneometer were used to measure transepidermal water loss (TEWL) and skin hydration in inner forearm. (f) The O2C-system (Lea Instruments) to assess peripheral blood flow and oxygen was used.	Skin elasticity and density hydration dermal blood flow oxygen saturation	roughness, volume, and scaling skin redness	skin thinning	Regular intake of tea flavonoid in beverage was able to protect skin against UV radiation and improve skin quality and function
Bog blueberry extract (BBe) [21]	Human dermal fibroblast	Human dermal fibroblasts were pretreated with 1–10 mg/L ATH-BBe. Then, 100 mJ/cm^2^ UV-B was used to expose the pre-treated cell culture. (a) The MTT assay was used to assess cellular viability following UVB exposure. (b) Procollagen and collagenolytic MMP secretion expression levels were assessed using Western blot. (c) ELISA was used to measure the cytokine secretion from UV-B-irradiated fibroblasts. (d) Immunoblot analysis was used to assess NF-kB activation and MAPK signalling.	Cell viability Procollagen expression	Expression of MMP-1, MMP-8, and MMP-13 Cytokine releaseꓽ IL-1b, IL-6, IL-8, TNF-a. Phosphorylation of IkB		Anthocyanins were able to alleviate UV radiation injury via anti-inflammatory action and inhibition of NF-kB activation and MAPK signalling.
Rosemary extract, citrus bioflavonoid extract [22]	In vitro study The human keratinocytes cell (HaCaT cells)	Human keratinocyte cells were cultured and treated with rosemary extract and citrus bioflavonoid extract before exposure to UVB. (a) Cell protection level was estimated as the percentage of cell viability recovered using the MTT assay, which was used to determine cellular viability. (b) The Folin-Ciocalteu technique was used to calculate the total phenolic compound (TPC). (c) The Comet assay was used to measure DNA damage. (d) A CBMN technique was used to conduct micronucleus assays on the irradiated lymphocytes.	Cell survival MED	Abnormalities of lymphocytes chromosomal Intracellular ROS DNA damage		Synergistic effect of both polyphenols in photoprotection by reducing oxidative damage.
	Human study Double blind, placebo controlled clinical trial	MED was measured pre and post-treatment.				
Luteolin from carrots [23]	In vitro study HaCaT human keratinocyte cell line	Before being exposed to UVB (0.01 J/cm^2^), HaCaT cells were treated for 1 h with luteolin (5 and 10 lM) and then incubated for 5 or 3 h, respectively. (a) The expression of MMP-1, JNK, c-Jun, ERK, and RSK phosphorylation were measured by Western blot analysis. (b) Using the luciferase assay, AP-1 transactivation and c-Fos promoter activation were quantified. (c) The expression of JNK1, JNK2, ERK2, and p90RSK was assessed using kinase assay. (d) JNK1, JNK2, and RSK2 expressions were analysed using pull-down assay.		MMP-1 expression Inhibit AP-1 transactivation C-FOS gene expression Activity of JNK1 and p90RSK Expression of MMp-13 Wrinkles formation	Activation of JNK2 and ERK	Luteolin was able to provide protection by inhibiting UVB-induced signal transduction
	In vivo study SKH-1 hairless mice (5 weeks of age; mean body weight, 25 g)	For 15 weeks, luteolin (10 and 40 nmol) dissolved in 200L acetone was applied topically to the backs of SKH-1 hairless mice. This treatment began 1 h before UVB exposure (0.18 J/cm^2^). (a) Western blot analysis was done to assess MMP-13 expression (b) Grading of wrinkle formation.				
Tomato and rosemary extract [24]	Double blind, placebo controlled clinical trial	149 participants were randomly assigned to carotenoid-rich tomato nutrient complex (TNC) or placebo for a 12-week period (a) A dose of 1.25 MED with Dermalight^®^ 80 MED was administered to an additional field (12 × 12 mm) of the buttock to take biopsies and to determine the skin’s colour using chromametry. A change in skin tone quantified the development of erythema. (b) Blood samples were taken, and serum was evaluated for the presence of carotenoids at random intervals during a 5-week washout phase, after four weeks of therapy, and after the study. Two colourless carotenoids (phytofluene and phytoene), were also measured. (c) Biopsies were taken after 24 h following chromametry using 1.25 MED in order to evaluate UV-induced gene expression.	Carotenoid plasma level	Erythema formation IL6	MED	The carotenoid-rich supplement significantly prevented UVB-induced erythema and cytokine upregulation.
Tomato [25]	Single blind, placebo controlled clinical trial.	Volunteers received either tomato paste and olive oil or just olive oil. Their skin was exposed to UVB and they were assessed for erythema at baseline and week 12. (a) Immunohistochemical analysis was done to identify a panel of ECM molecules or remodelling enzymes in frozen sections. (b) A five-point semiquantitative scale was used to evaluate the level of immunostaining for procollagen and fibrillin-1, (c) MMP-1-positive dermal cell counts were calculated with high-power field (×400)	Mean SD erythemal D30 Procollagen deposition	MMP-1 expression	Fibrillin-1 expression	Lycopene-rich tomato paste offers defense against both immediate and possible long-term effects of photodamage.
Grape peel extract (GPE) [26]	In vitro study HaCaT cells In vivo study: ICR six-week-old male mice	HaCaT cells were treated for 24 h with GPE or resveratrol (RES). Then, HaCaT cells were subsequently exposed to UVB radiation at a dose of 25 mJ/cm^2^. (a) Assessment of fractionation of nuclear and cytoplasmic proteins using Western blot analysis. Before receiving UVB (280–315 nm) radiation treatment, the dorsal skin of mice was shaved and they were given GPE orally three times per week for seven weeks. (a) Histological analysis of the skin tissue from mice (b) Western blot analysis	Expression of nuclear Nrf2 Expression of cytoplasmic HO-1	Wrinkles Epidermis thickening		Resveratrol exhibits photoprotection and photoageing effects via regulation of nuclear Nrf2.
Grape seed proanthocyanidin extract (GSPE) [27]	Double blind, placebo controlled clinical trial.	Cutaneous areas on the backs of volunteers were untreated or treated with GSPE solutions, 30 min before exposure to two minimal erythema doses (MED) of solar simulated radiation. (a) Cutaneous regions at various sites were examined histologically for the number of sunburn cells. (b) Mutant p53-positive cells and CD1a+ Langerhans cells were identified using immunohistochemistry.	Langerhans cells exhibit more dendrites	Sunburn cells Fewer mutant p53-positive cells		Topical GSPE prevents formation of sunburn cells, mutant p53 epidermal cells.

## Data Availability

This manuscript does not contain original data.

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
