# Peer review of "A Scoping Review on the Effects of Carotenoids and Flavonoids on Skin Damage Due to Ultraviolet Radiation"

_nutrients, 2022, doi:10.3390/nu15010092_

Round 1

Reviewer 1 Report

This review article summarizes the protective role of carotenoids and flavonoids in the skin. the article is well-written with adequate references. However, some references show "error" in the citations. Please correct these errors in the links. 

Author Response

Thank you very much for your comments. We have made the necessary corrections in the revised manuscript.

Reviewer 2 Report

It seemed to search for papers that fit the criteria and just introduce them. This content lacks the impact that would lead to acceptance and is not appropriate for this journal. In order for this manuscript to be accepted, it is necessary to make a very large revision.

1. There are some ambiguities in the flowchart in this manuscript. The following should be confirmed and clarified or corrected.

1-1. How did 71 articles remain from the 431 searched articles? The description that 162 articles were excluded does not match the balance.

1-2. After the second phase, 21 out of 71 articles were excluded, then 42 articles remained also does not match the balance.

2. This manuscript aims to conduct a systematic review of the effects of flavonoids and carotenoids on UV-exposed skin. However, the results obtained are 9 articles, and I feel it is insufficient to be called systematic. It might be better to broaden the scope and include more literature, as the discussion has more articles.

Following, it is to be needed some minor revisions.

1. Since the table precedes the discussion, the citation numbers of the papers shown in the table should be 19-27. Or it is suggested that split the table into flavonoid studies and carotenoid studies.

2. There are various definitions of the word “superfood” that differ in the food groups they contain. As such, the term is unscientific and inappropriate for use in such papers.

Round 2

Reviewer 2 Report

It has been improved from before the revision, and there is no problem even if it is accepted. Please revise the following minor points.

 ãƒ»The citation number of the reference is not appropriate. In particular, it is written as "0" in the introduction. Please double-check the whole thing.